# CXCR6+ Tumor-Associated Macrophages Identify Immunosuppressive Colon Cancer Patients with Poor Prognosis but Favorable Response to Adjuvant Chemotherapy

**DOI:** 10.3390/cancers14194646

**Published:** 2022-09-24

**Authors:** Jiang Chang, Songbin Lin, Yihao Mao, Yuqiu Xu, Zhiyuan Zhang, Qi Wu, Yijiao Chen, Ye Wei, Qingyang Feng, Jianmin Xu

**Affiliations:** 1Colorectal Cancer Center, Zhongshan Hospital, Fudan University, Shanghai 200000, China; 2Department of General Surgery, Zhongshan Hospital, Fudan University, Shanghai 200000, China; 3General Surgery Department, Zhongshan Hospital, Fudan University (Xiamen Branch), Xiamen 361000, China; 4Shanghai Engineering Research Center of Colorectal Cancer Minimally Invasive, Shanghai 200000, China

**Keywords:** tumor-associated macrophages, colon cancer, adjuvant chemotherapy, immune microenvironment

## Abstract

**Simple Summary:**

In this study, we first reported the infiltration and prognostic value of CXCR6+TAMs in all stages of colon cancer patients and assessed predictive ability as a biomarker for different duration adjuvant chemotherapy regimens in the primary cohort and validation cohort Patients with high CXCR6+ TAM infiltration tended to have worse overall survival. A 6-month chemotherapy regimen for high-risk stage II and stage III patients after curative operation benefited those with high CXCR6+ TAM density the most. Furthermore, we found there was a negative relationship between CXCR6+TAMs and activated CD8+ T cells. Cytokines from peripheral blood reflected the immunosuppressive state in patients with high accumulations of CXCR6+TAMs This indicates that CXCR6+TAMs may be involved in the formation of an immunosuppressive environment.

**Abstract:**

We explored the infiltration and prognostic value of CXCR6+TAMs in all stages of colon cancer (CC) patients and assessed predictive ability as a biomarker for different ACT regimens among high-risk stage II and stage III patients in both primary and validation cohorts. Two independent cohorts of 360 and 126 consecutive colon cancer patients were enrolled from two medical centers of Zhongshan Hospital. Immunofluorescence and immunohistochemistry were performed to detect the density of CXCR6+TAMs and activated CD8+ T cells. The infiltration of CXCR6+TAMs was higher in tumor tissues and increased with advanced tumor stage. A high density of CXCR6+TAMs predicted worse overall survival (OS) in all CC patients (HR = 2.49, 95% CI = (1.68, 3.70), *p* < 0.001), and was an independent risk factor verified by Cox regression analysis (HR = 1.68, 95% CI = (1.09, 2.59), *p* = 0.019). For high-risk stage II and stage III patients with a high density of CXCR6+TAMs, better disease-free survival (DFS) (HR = 0.32, 95% CI = (0.11, 0.89), *p* = 0.003), and OS (HR = 0.28, 95% CI = (0.07, 1.11), *p* = 0.014) were observed in the 6-month treatment group. There was a negative relationship between the density of CXCR6+TAMs and CD8+ T cells (R = −0.51, *p* < 0.001) as well as activated CD8+ T cells (R = −0.54, *p* < 0.001). Higher levels of IL-6 and lower levels of IL-2R and TNF-α were expressed in high-CXCR6+ TAM-density patients, which indicates that CXCR6+TAMs contribute to an immunosuppressive microenvironment. CXCR6+TAMs predicted prognosis and response to different durations of ACT in CC patients. CXCR6+TAMs were associated with an immunosuppressive microenvironment and suppressed the activation of CD8+ T cells.

## 1. Introduction

Colon cancer (CC) is one of the leading causes of cancer mortality all over the world [1], and the majority are stage II and stage III diseases. With reference to current guidelines, high-risk stage II and all stage III patients should receive adjuvant chemotherapy (ACT) after surgery. However, they do not respond equally [2,3,4], and considering the cumulative sensory neurotoxicity of oxaliplatin, the duration of therapy is still full of controversy. Based on the results of IDEA [5,6], although less toxicity was observed in the 3-month treatment, noninferiority was not demonstrated when compared with the 6-month treatment for the overall study population. So other biomarkers to better identify the patients who are appropriate for combination chemotherapy are needed.

With the development of research such as Immunoscore and KEYNOTE-177 in recent years [7,8], the tumor immune microenvironment has been placed in a position that can no longer be ignored. Tumor-promoting inflammation is recognized as one of the hallmarks of cancer [9]. Tumor-associated macrophages (TAMs), which are macrophages residing in tumor tissues, constitute the majority of immune cells infiltrating the tumor immune microenvironment to regulate immunological surveillance and participate in the process of tumor growth, invasion, metastasis, and drug resistance [10,11]. The accumulation of TAMs is reported to be related to many kinds of chemokines, by which circulating monocytes are recruited and differentiated into TAMs [12,13]. TAMs may be activated classically (known as the M1 type) by IFN-γ and TNF to suppress tumor progression, or alternatively (known as the M2 type) by IL-4 and IL-10 to promote tumor immune evasion in different conditions [14]. Our previous study demonstrated M2-polarized TAMs could act as prognostic and predictive biomarkers for postoperative adjuvant chemotherapy in patients with stage II colon cancer [15]. However, emerging studies revealed that the TME is critically complicated, and macrophages may go beyond the typical M1/M2 subsets according to their functions [16]. TAMs with assorted molecular signatures still need to be explored.

C-X-C motif chemokine receptor 6 (CXCR6), also known as STRL33 and TYMSTR [17], is a G-protein-coupled receptor highly expressed in the thymus, appendix, placenta, spleen, and lymph nodes [18]. In the immune system, CXCR6 is selectively expressed by CD4+ T cells, CD8+ T cells [19], natural killer (NK) cells [20], and plasma cells [21], and is mainly attracted by CXC Motif Chemokine Ligand 16 (CXCL16) [22]. In cancer, CXCR6 was demonstrated to be related to greater cancer cell invasiveness in prostate cancer [23], while inhibiting proliferation and migration in renal cancer [24]. A lower expression of CXCR6 was found in colorectal cancerous tissue compared to normal tissue [25]. However, the expression and the role of CXCR6-positive macrophages has rarely been explored. It has been reported that CXCR6 shows the ability to promote M2 polarization in macrophages at the maternal–fetal interface [26]. Higher CXCR6 expression in macrophages stimulated with specific cytokines was also observed in some in vitro experiments [27,28]. But the function of CXCR6+TAMs in the tumor immune microenvironment is still unknown.

In this study, we explored the infiltration and prognostic value of CXCR6+TAMs in all stages of CC patients and assessed predictive ability as a biomarker for different ACT regimens among high-risk stage II and stage III patients in both primary and validation cohorts. Furthermore, we also analyzed the correlation between CXCR6+TAMs and activated CD8+ T cells, and cytokines from peripheral blood to explore the influence on the immune microenvironment.

## 2. Materials and Methods

### 2.1. Patient Population

Two independent patient cohorts were recruited in this study. A total of 360 consecutive CC patients who underwent radical colectomy between 2009.09 and 2011.12 from Zhongshan Hospital, Fudan University (Shanghai, China) were included retrospectively as the primary cohort, and 126 CC patients were enrolled from Zhongshan Hospital, Fudan University (Xiamen, China), between 2016.06 and 2019.08 as the validation cohort. 

The inclusion criteria were as follows: (a) age ranges from 18 to 80 years old, (b) received radical (R0) resection (both primary tumor and metastases for stage IV patients), (c) pathologically diagnosed as CC according to the International Union Against Cancer (UICC)/American Joint Committee on Cancer (AJCC) TNM staging system 9th edition. The exclusion criteria were: (a) emergency surgery including bowel obstruction and perforation, (b) hereditary colorectal cancer (including familial adenomatous polyposis, Lynch syndrome, and MYH-associated polyposis), (c) multiple primary tumors, (d) tissue specimen, or unavailable clinicopathological data. The definition of CC was a tumor localized 15 cm above the anal verge. The overall survival (OS) was defined as the time between the date of surgery to the date of death, and disease-free survival (DFS) was calculated from the date of surgery to the date of the first tumor recurrence. In stage II CC, T3N0M0 with at least one of the factors (poorly differentiated histology, vascular invasion, perineural invasion, <12 lymph nodes examined) or T4N0M0 (pMMR) was recognized as high-risk. Adjuvant chemotherapy (FOLFOX or CAPOX) was applied through comprehensive consideration of NCCN guidelines (version 4.2021), treatment tolerance, and patient preference.

The clinicopathological information and expression of cytokines from peripheral blood before surgery including IL-2 receptor (IL-2R), IL-6, and TNF-α were recorded from the database of the hospital retrospectively. Written informed consent was acquired from every patient, and this study was approved by the Clinical Research Ethics Committee of Zhongshan Hospital, Fudan University.

### 2.2. Immunohistochemistry (IHC) and Immunofluorescence (IF)

Formalin-fixed and paraffin-embedded tissues of CC were collected. IHC was applied to detect the expression of CD8+ T cells and activated CD8+ T cells, and the procedure was described in detail previously [15]. Briefly, tissue sections were incubated with the primary antibody CD8 (Rabbit, 1:200, ab237709, abcam) and GZMB (Rabbit, 1:250, ab134933, abcam), respectively, in a moist chamber overnight at 4 °C. DAB reagent was used for staining. IF was carried out to detect the expression of CXCR6+TAMs, and the sections were incubated with the primary antibody CXCR6 (Rabbit, 1:500, ab237116, abcam) and CD68 (Mouse, 1:1000, ab237116, abcam) overnight at 4 °C. Then the sections were incubated with FITC- and TRITC-conjugated secondary antibodies for 2 hat room temperature. Finally, the slides were mounted with Antifade Mounting solution containing DAPI. The slides were analyzed using Nikon Eclipse C1. Two pathologists, blinded to patient clinical and follow-up data, recorded the density of immune cells as the mean number of cells/mm² under a high-power field (HPF) of 200× of each specimen from five randomized fields independently. If the difference of density was greater than ten percent, a recount was performed. The ratio of the counts of GZMB to CD8 was defined as the percentage of activated CD8+ T cells [29]. 

### 2.3. Statistical Analysis

Statistical analysis was performed by R project (version 4.1.2). Category variables were compared using Pearson’s chi-squared test, chi-square with continuity correction, and Fisher’s exact test when appropriate. The Kaplan−Meier method with a log-rank test was applied to compare OS and DFS. Clinicopathological variables first filtrated by univariate cox regression (*p* < 0.10), were then analyzed using multivariate cox analysis, and a two-tailed *p* value < 0.05 was identified as statistically significant.

## 3. Results

### 3.1. CXCR6+TAMs Infiltrated CC Tissues and Predict Poor Prognosis in All Stages Patients

Double immunofluorescence staining was performed in tissue sections of both primary and validation cohorts to reveal the colocalization of CXCR6 and macrophage marker CD68. As shown, CXCR6+ macrophages were a subset of TAMs in CC (Figure 1A). Compared with peritumor tissues, CXCR6+TAMs infiltrated more in tumor tissues (Figure 1B). Furthermore, we analyzed the relationship between clinicopathological information and CXCR6+TAMs (Table 1); the median value (56 per mm²) was chosen as the cut-off point for CXCR6+TAM high- and low-density groups. The expression of CXCR6+TAMs was higher in lymph node metastasis and distant metastasis groups. The baseline data are summarized in Appendix A, and the infiltration of CXCR6+TAMs in peritumor tissue is shown in Appendix A.

To explore the potential impact of CXCR6+TAMs on clinical prognosis, we conducted the Kaplan−Meier analysis of CXCR6+TAMs in all patient stages (Figure 1C). Better OS was observed in the high CXCR6+ TAM infiltration group in the primary cohort and validation cohort (*p* < 0.001 and *p* = 0.004). Univariate and multivariate cox regression was constructed, and CXCR6+TAMs were also confirmed as an independent prognostic factor for OS both in the primary cohort (hazard ratio (HR): 1.68, 95% confidence interval (CI): 1.09–2.59, *p* = 0.019) (Table 2) and validation cohort (HR: 2.07, 95% CI: 1.03–4.17, *p* = 0.041) (Appendix A). These findings imply that CXCR6+TAMs were positively related to tumor progression and poor prognosis in CC.

### 3.2. CXCR6+TAMs Predict the Benefit of 6-Month Adjuvant Chemotherapy in High-Risk Stage II and Stage III Patients 

ACT is applied routinely for high-risk stage II and stage III patients after radical surgery, but the choice of 3-month and 6-month therapy is still full of controversy, so we also assessed the predictive ability of CXCR6+TAMs as a biomarker for chemotherapy duration. Interestingly, we found DFS of patients with high CXCR6+ TAM infiltration was significantly prolonged by the 6-month ACT in the primary cohort (*p* = 0.003) and validation cohort (*p* =0.039), while in patients with low CXCR6+TAMs, there was no infiltration (*p* = 0.847, primary cohort; *p* = 0.670, validation cohort) (Figure 2A,B). Similarly, the OS of patients with high CXCR6+ TAM infiltration was also prolonged by the 6-month ACT in the primary cohort (*p* = 0.014) and validation cohort (*p* = 0.048), while in patients with low CXCR6+TAMs, there was no infiltration (*p* = 0.850, primary cohort; *p* = 0.826, validation cohort) (Appendix A). Subgroup analysis revealed that 6 months of chemotherapy benefited male, mucinous, tumor size ≤ 4 cm, pMMR especially (Figure 3). Cumulatively, these results suggest that the infiltration CXCR6+TAMs could predict the benefit of 6-month adjuvant chemotherapy in high-risk stage II and stage III patients (Table 3).

### 3.3. CXCR6+TAMs Contribute to Immunosuppressive Microenvironment 

To reveal how CXCR6+TAMs influence the immune microenvironment, we further analyzed the CD8+ T cell infiltration in CC by IHC (Figure 4A). More CD8+ T cells and activated CD8+ T cells were observed in tissues with a low density of CXCR6+ TAM. In all patient stages, there was a negative relationship between the number of CD8+ T cells and CXCR6+ TAM (R² = 0.260, *p* < 0.001) (Figure 3B), and the same was seen for activated CD8+ T cells (R² = 0.292, *p* < 0.001) (Figure 4C). Higher levels of IL-2R (*p* < 0.001), TNF-α (*p* = 0.021) and lower levels of IL-6 (*p* = 0.003) (Figure 4D) were detected in low-CXCR6+TAM-density patients. The results above indicate that CXCR6+TAMs might contribute to an immunosuppressive microenvironment by suppressing the activation of CD8+ T cells.

## 4. Discussion

Tumor-promoting inflammation was recognized as one of the hallmarks of cancer [9]. As a cancer of high heterogeneity, the immune microenvironment in colon cancer still needs to be explored. Tumor-associated macrophages constitute the majority of immune cells [10]. Previous studies suggested that TAMs with different molecular signatures could be identified as prognostic biomarkers and therapeutic targets, so it is meaningful to research new subtypes.

CXCR6 is one of the chemokine receptors expressed in immune cells. The expression of CXCR6 in human tumor tissues is closely related to the abundance of cytotoxic T lymphocyte (CTL), and it inhibits tumor growth by maintaining the survival of CTL, which is the most important among all chemokine receptors [30]. However, the role of CXCR6 on macrophages has rarely been reported. At the maternal–fetal interface, CXCL16 secreted by a trophoblast stimulates the CXCR6 macrophage receptor, which can down-regulate CD80, CD86, and HLA-DR, while up-regulating the expression of IL-10 and CD163, resulting in M2 polarization [26]. In tumors, although CXCL16 could mediate the anti-inflammatory phenotype of microglia (brain-resident macrophages), the expression of CXCR6 in these TAMs was still relatively low [27,31]. On the other hand, some in vitro data showed that macrophages stimulated by M-CSF or GM-CSF tended to express higher CXCR6 [27,28]. For this reason, the expression and significance of CXCR6 on TAMs still needs to be explored further. In this article, we first reported that CXCR6 was expressed by macrophages in CC, which could also be positively related to tumor progression, thus predicting a worse prognosis.

Combination adjuvant chemotherapy should be routinely applied to high-risk stage II and all stage III patients after surgery according to guidelines. However, about 15% of stage II patients still suffer tumor recurrence and progression [2], and stage III patients do not respond to chemotherapy equally [3,4,32]. Considering the cumulative side effects, it is still meaningful to find biomarkers to identify patients who are more suitable for the 6-month regimen. Interestingly, we found that patients with high infiltration of CXCR6+ macrophages tended to have a poor prognosis, but a better response to 6-month chemotherapy. Although chemotherapy-related toxicity was also duration-dependent, the survival benefit still gives reason to believe that patients with high CXCR6+TAM density are more suitable candidates for 6 months of chemotherapy. Therefore, our results indicate that CXCR6+TAMs could be a potential prognostic and predictive biomarker for patients with colon cancer. Some specific patterns of immune cell infiltration in the microenvironment can reflect specific clinical meaning. Previous reports showed that the CD8+ T cells in colorectal cancer could be of prognostic value [33]. The results of IHC indicate that there was a negative relationship between the infiltration of activated CD8+ T cells and CXCR6+TAMs. Soluble cytokines in peripheral blood can reflect the status of some immune cells, and IL-2R, IL-6, and TNF-α were the most commonly detected. The proliferation of T lymphocytes is triggered by the interaction of IL-2, and the level of IL-2R reflects the proliferation of T cells [34]. The IL-6/IL-6R signaling axis promotes M2 macrophage polarization [35], and TNF-α was mainly secreted by M1 macrophage to play a proinflammatory response [36]. Based on the results above, we believe that CXCR6+TAMs contribute to immunosuppressive microenvironments.

## 5. Conclusions

In summary, we first report that CXCR6 could be expressed by TAMs, and CXCR6+TAMs can serve as a prognostic and chemotherapy duration biomarker in CC. CXCR6+TAMs may play a pivotal role in suppressing tumor immunity by suppressing the activation of CD8+ T cells.

## Figures and Tables

**Figure 1 cancers-14-04646-f001:**
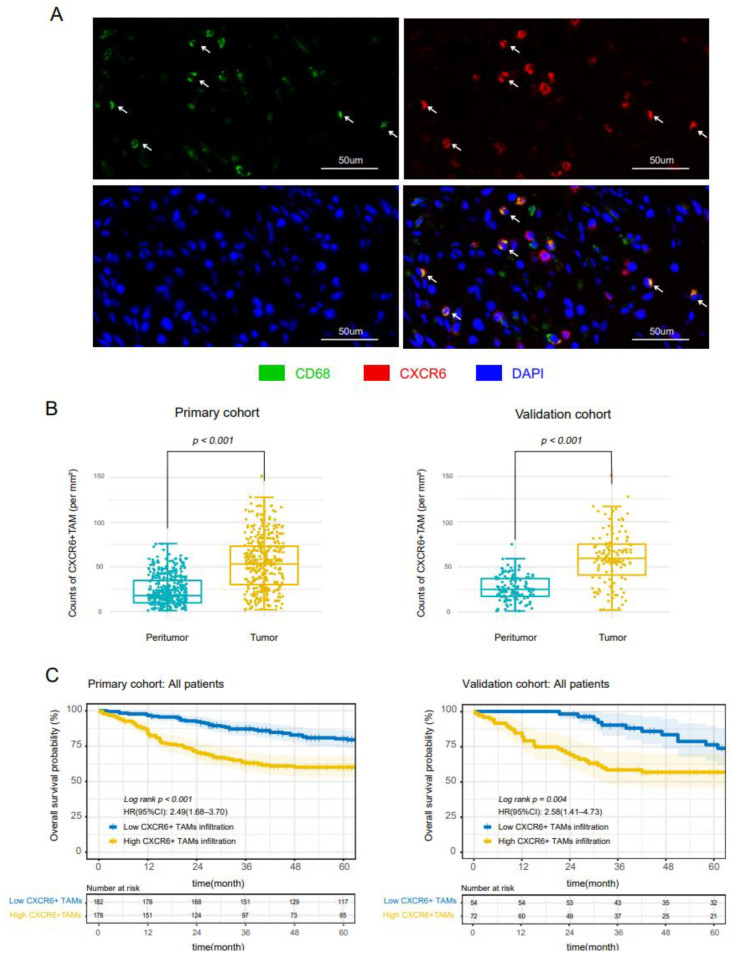
**CXCR6+TAMs are infiltrated in CC and predict poor prognosis in all patient stages.** (**A**) Representative images of CXCR6+TAMs in CC specimens. White arrows indicate CXCR6+TAMs. (**B**) Statistical analysis of the density of CXCR6+TAMs in tumor tissues and peritumor tissues in two independent cohorts. (**C**) Overall survival curves of all patients with high and low CXCR6+ TAM infiltration in two independent cohorts.

**Figure 2 cancers-14-04646-f002:**
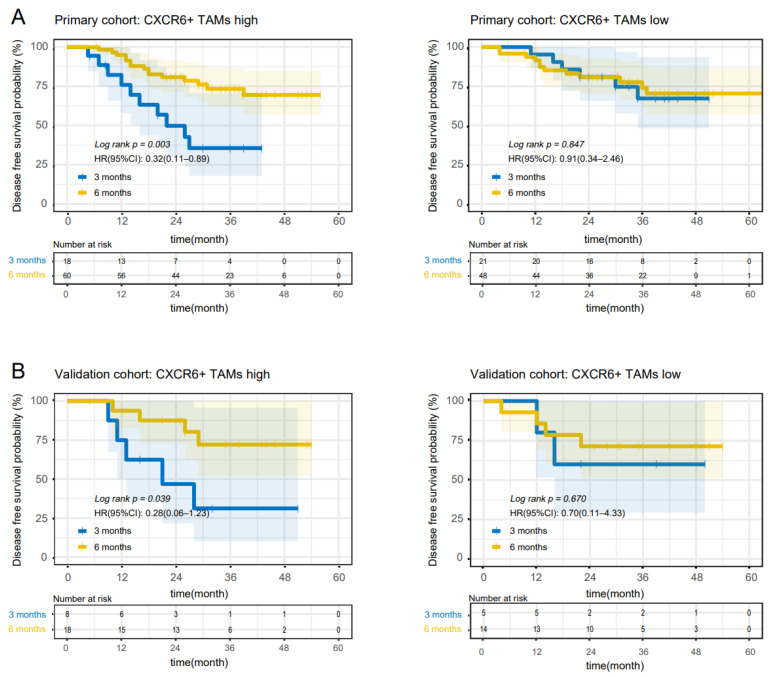
High CXCR6+ TAM infiltration is associated with a better response to 6-month adjuvant chemotherapy in high-risk stage II and stage III CC patients. (**A**) Disease-free survival curves of patients with high and low CXCR6+TAMs from the primary cohort. (**B**) Disease-free survival curves of patients with high and low CXCR6+TAMs from the validation cohort.

**Figure 3 cancers-14-04646-f003:**
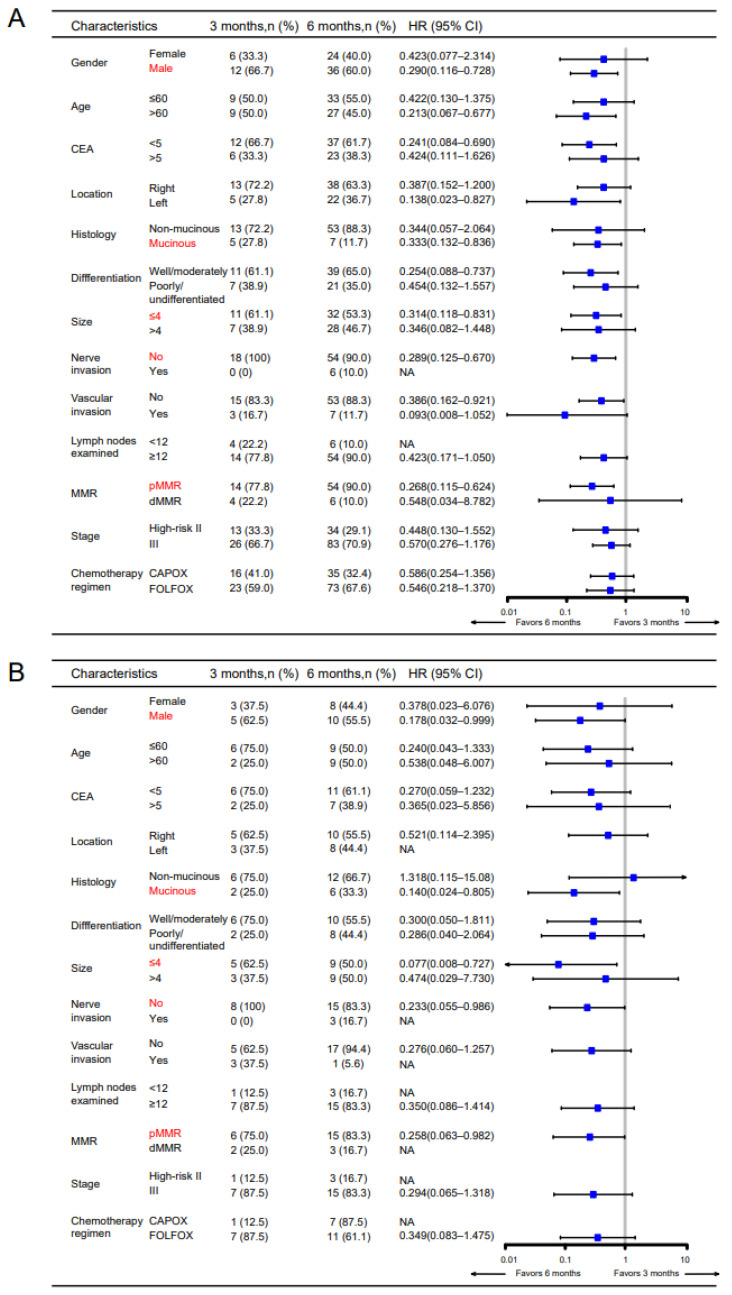
**Subgroup analysis of high-risk stage II and stage III CC patients receiving adjuvant chemotherapy.** Forest plot for primary cohort (**A**) and validation cohort (**B**). Groups with statistical significance for both cohorts were marked.

**Figure 4 cancers-14-04646-f004:**
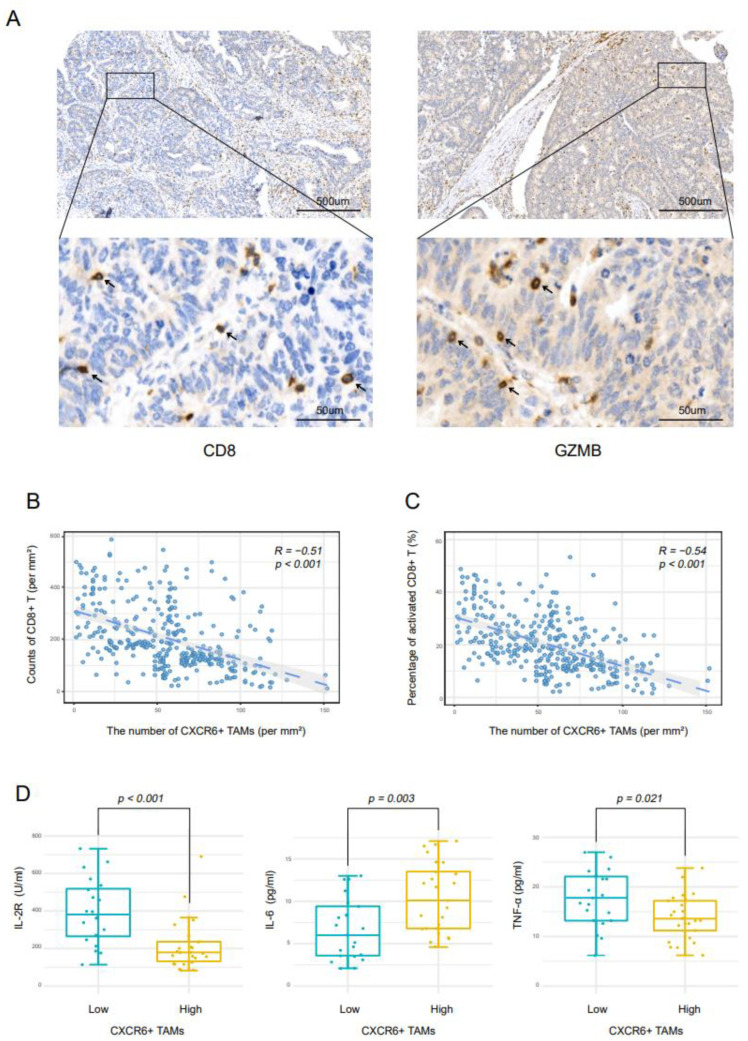
CXCR6+TAMs contribute to an immunosuppressive microenvironment. (**A**) Representative images of CD8+ T cells and GZMB+ cells in CC specimens. (**B**) Correlation between CXCR6+TAMs and CD8+ T cells. (**C**) Correlation between CXCR6+TAMs and activated CD8+ T cells. (**D**) Statistical analysis of the levels of cytokines in CC patients with high and low CXCR6+TAMs.

**Table 1 cancers-14-04646-t001:** Relationship between the number of CXCR6+TAMs and clinical characteristics.

	CXCR6+TAMs Primary Cohort			CXCR6+TAMs Validation Cohort		
	Low (%)	High (%)	P	Low (%)	High (%)	P
**All patients**	182 (50.6)	178 (49.4)		54 (42.9)	72 (57.1)	
**Gender**			0.361			0.344
Female	75 (41.2)	65 (36.5)		24 (44.4)	26 (36.1)	
Male	107 (58.8)	113 (62.5)		30 (55.6)	46 (63.9)	
**Age**			0.911			0.439
≤60	93 (51.1)	92 (51.7)		27 (50.0)	41 (56.9)	
>60	89 (48.9)	86 (48.3)		27 (50.0)	31 (43.1)	
**CEA (ng/mL)**			**0.021**			0.797
≤5	109 (59.9)	85 (47.8)		29 (53.7)	37 (51.4)	
>5	73 (40.1)	93 (52.2)		25 (46.3)	35 (48.6)	
**Tumor location**			0.263			0.165
Right-sided colon	104 (57.1)	112 (62.9)		24 (44.4)	41 (56.9)	
Left-sided colon	78 (42.9)	66 (37.1)		30 (55.6)	31 (43.1)	
**Tumor size (cm)**			0.850			0.322
≤4	102 (56.0)	98 (55.1)		20 (37.0)	33 (45.8)	
>4	80 (44.0)	80 (44.9)		34 (63.0)	39 (54.2)	
**Histology**			0.582			0.183
Non-mucinous	157 (86.3)	157 (88.2)		47 (87.0)	56 (77.8)	
Mucinous	25 (13.7)	21 (11.8)		7 (13.0)	16 (22.2)	
**Differentiation**			**0.018**			0.380
Well/moderately	141 (77.5)	118 (66.3)		42 (77.8)	51 (70.8)	
Poorly/undifferentiated	41 (22.5)	60 (33.7)		12 (22.2)	21 (29.2)	
**T stage**			0.720			0.532
T1–2	77 (42.3)	72 (40.4)		21 (38.9)	32 (44.4)	
T3–4	105 (57.7)	106 (59.6)		33 (61.1)	40 (55.6)	
**N stage**			**<0.001**			**0.006**
N0	118 (64.8)	61 (34.3)		35 (64.8)	29 (40.3)	
N1–2	64 (35.2)	117 (65.7)		19 (35.2)	43 (59.7)	
**M stage**			**<0.001**			**0.027**
M0	153 (84.1)	110 (61.8)		41 (75.9)	41 (56.9)	
M1	29 (15.9)	68 (38.2)		13 (24.1)	31 (43.1)	
**TNM stage**			**<0.001**			0.104
I	41 (22.5)	14 (7.8)		11 (20.4)	9 (12.5)	
II	67 (36.8)	32 (18.0)		15 (27.8)	12 (16.7)	
III	45 (24.7)	64 (36.0)		15 (27.8)	20 (27.8)	
IV	29 (15.9)	68 (38.2)		13 (24.1)	31 (43.1)	
**Nerve invasion**			0.156			0.998
No	171 (94.0)	160 (89.9)		51 (94.4)	67 (93.1)	
Yes	11 (6.0)	18 (10.1)		3 (5.6)	5 (6.9)	
**Vascular invasion**			**0.017**			0.643
No	163 (89.6)	148 (83.1)		48 (88.9)	62 (86.1)	
Yes	19 (10.4)	30 (16.9)		6 (11.1)	10 (13.9)	
**Lymph nodes examined**			0.156			0.803
<12	11 (6.0)	18 (10.1)		3 (5.6)	6 (8.3)	
≥12	171 (94.0)	160 (89.9)		51 (94.4)	66 (91.7)	
**MMR status**			0.357			0.122
dMMR	17 (9.3)	22 (12.4)		4 (7.4)	12 (16.7)	
pMMR	165 (90.7)	156 (87.6)		50 (92.6)	60 (83.3)	

CEA, carcinoembryonic antigen; MMR, mismatch repair.

**Table 2 cancers-14-04646-t002:** Cox regression analysis for OS of primary cohort.

	Univariate		Multivariate	
	HR (95% CI)	P	HR (95% CI)	P
**Gender**				
Female	1 (reference)			
Male	1.15 (0.77–1.72)	0.499		
**Age**				
≤60	1 (reference)			
>60	0.94 (0.63–1.31)	0.749		
**Preoperative serum CEA (ng/mL)**				
≤5	1 (reference)		1 (reference)	
>5	2.91 (1.92–4.40)	**<0.001**	1.80 (1.16–2.81)	**0.009**
**Primary tumor location**				
Right-sided colon	1 (reference)			
Left-sided colon	0.72 (0.48–1.09)	0.124		
**Primary tumor size (cm)**				
≤4	1 (reference)			
>4	1.18 (0.80–1.74)	0.405		
**Histology**				
Non-mucinous	1 (reference)			
Mucinous	0.96 (0.62–1.47)	0.839		
**Primary Differentiation**				
Well/moderately	1 (reference)			
Poorly/undifferentiated	1.40 (0.92–2.11)	0.113		
**T stage**				
T1–2	1 (reference)		1 (reference)	
T3–4	0.70 (0.47–1.03)	**0.070**	0.78 (0.52–1.16)	0.212
**N stage**				
N0	1 (reference)		1 (reference)	
N1–2	2.01 (1.33–3.00)	**0.001**	1.06 (0.69–1.63)	0.788
**M stage**				
M0	1 (reference)		1 (reference)	
M1	4.17 (3.12–5.56)	**<0.001**	3.30 (2.41–4.53)	**<0.001**
**TNM stage**				
I	1 (reference)			
II	2.01 (0.74–5.44)	0.171		
III	1.83 (0.67–4.95)	0.236		
IV	12.89 (5.16–32.20)	**<0.001**		
**Nerve invasion**				
No	1 (reference)		1 (reference)	
Yes	1.79 (0.96–3.35)	**0.068**	0.96 (0.51–1.81)	0.894
**Vascular invasion**				
No	1 (reference)			
Yes	1.34 (0.79–2.25)	0.274		
**Lymph nodes examined**				
<12	1 (reference)			
≥12	0.88 (0.43–1.82)	0.739		
**MMR status**				
pMMR	1 (reference)		1 (reference)	
dMMR	1.72 (1.01–2.93)	**0.048**	1.45 (0.85–2.50)	0.177
**The density of CXCR6+TAMs**				
Low	1 (reference)		1 (reference)	
High	2.51 (1.66–3.79)	**<0.001**	1.68 (1.09–2.59)	**0.019**

CEA, carcinoembryonic antigen; MMR, mismatch repair.

**Table 3 cancers-14-04646-t003:** Results of CXCR6+TAMs as a biomarker.

All Patient Stages: High Density vs. Low Density of CXCR6+TAMs						
**Cohort**	**No. of Patients (%)**	**OS**				
		**HR (95%)**	***p*** **Value**			
Primary	360 (100)	2.49 (1.68–3.70)	<0.001			
Validation	126 (100)	2.58 (1.41–4.73)	0.004			
**High-risk stage II and stage III patients: 6 months vs. 3 months of FOLFOX/CAPOX**						
Cohort	**CXCR6+TAM infiltration**	**No. of patients (%)**	**DFS**		**OS**	
			**HR (95%)**	***p*** **value**	**HR (95%)**	***p*** **value**
Primary	High	78 (53.1)	0.32 (0.11–0.89)	0.003	0.28 (0.07–1.11)	0.014
	Low	69 (46.9)	0.91 (0.34–2.46)	0.847	0.89 (0.26–3.03)	0.850
Validation	High	26 (57.8)	0.28 (0.06–1.23)	0.039	0.29 (0.07–1.27)	0.048
	Low	19 (42.2)	0.70 (0.11–4.33)	0.670	0.83 (0.14–4.80)	0.826

## Data Availability

The datasets during and/or analyzed during the current study are available from the corresponding author on reasonable request.

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
