# Peer review of "CXCR6+ Tumor-Associated Macrophages Identify Immunosuppressive Colon Cancer Patients with Poor Prognosis but Favorable Response to Adjuvant Chemotherapy"

_cancers, 2022, doi:10.3390/cancers14194646_

Round 1
Reviewer 1 Report
The findings that high density of CXCR6+ TAM in CRC patients correlates with poor prognosis and better DFS and OS were observed in 6 months Adjuvant chemotherapy compare to 3 months treatment is important and may reflect the severity of the disease . A more detailed and explained tables would assist in better understanding the results.
Author Response
Thanks for you advice, and I totally agree that a more detailed and explained table is need. Results of CXCR6+TAMs as a biomarker have been summarized as the Table 3 in the article.
Reviewer 2 Report
The manuscript titled “CXCR6+ tumor-associated macrophages (TAM) identify immunosuppressive colon cancer patients with poor prognosis but favorable response to adjuvant chemotherapy by Chang et al., is to investigate the infiltration and prognostic value of CXCR6+ TAMs in all stages colon cancer (CC) patients, and assessed predictive ability as a biomarker for different adjuvant chemotherapy regimens among high-risk stage II and stage III patients both in primary and validation cohort. Overall, the objective of the work is not clear, the manuscript is poorly written with lots of spelling mistakes and seems to lack proofreading. The point-wise comments are as follows;
1. In the introduction, the TAMs are activated via M1 and M2 but what the TAMs are and how they infiltrate the tumors is not clear.
2. In the introduction it is also not clear how TAMs are associated with CXCR6.
3. What is the expression pattern of CXCR6 in the normal individual?
4. There are lots of spelling mistakes on page 2 “Tumor-promoting inflammation was recognized as “on” of the hallmarks of cancer” page 3 The overall survival (OS) was “defined” as the time between the date of surgery to the date of death, page 10 The results of IHC indicated that there was a negative relationship between the “infiltrattion” of activated and TNF-α was “maily” secreted by M1 macrophage to play Proin-flammatory response.
5. In figure 1A it is not clear what is really signal. Please increase the magnification and put arrows at the +ve signal and add a scale bar in each IFC and OHC image.
6. Figure 3 is not easy to understand font size to tiny.
7. Was any control sample like biopsy of a healthy colon, though it is not easy to find, but did the author use any control sample to normalize the CC values?
8. What is primary cohort and validation cohort? please elaborate it.
Reviewer 3 Report
This work entitled “CXCR6+ tumor-associated macrophages identify immunosuppressive colon cancer patients with poor prognosis but the favorable response to adjuvant chemotherapy” by Chang et al., They have reported that there is a negative relationship between the density of CXCR6+ TAMs and CD8+ T cells, as well as activated CD8+ T cells. The authors also reported that higher levels of IL-6, lower levels of IL-2R, and TNF-α were expressed in high CXCR6+ TAMs patients, which indicated that CXCR6+ TAMs contribute to an immunosuppressive microenvironment. It is professionally written, and well presented, but there are some minor comments:-
(1). Most importantly, in this study, the author didn’t check the CXCR6 level any of colon cancer. It might be important to check the expression pattern of this marker.
(2). Authors also quantified various cytokines level such as IL-6, TNF-alpha etc. They didn’t mention the rationale behind it.
(3). They use only Immunohistochemistry (IHC) and Immunofluorescence (IF), they should use other techniques like western blot to reform this expression.
(4). Discuss is very short, please add more information in the discussion section.
